# Characterization of the Cross-Species Transmission Potential for Porcine *Deltacoronaviruses* Expressing Sparrow Coronavirus Spike Protein in Commercial Poultry

**DOI:** 10.3390/v14061225

**Published:** 2022-06-05

**Authors:** Moyasar A. Alhamo, Patricia A. Boley, Mingde Liu, Xiaoyu Niu, Kush Kumar Yadav, Carolyn Lee, Linda J. Saif, Qiuhong Wang, Scott P. Kenney

**Affiliations:** 1Center for Food Animal Health, Department of Animal Sciences, College of Food, Agricultural and Environmental Sciences, The Ohio State University, Wooster, OH 44691, USA; maed@ucdavis.edu (M.A.A.); boley.28@osu.edu (P.A.B.); liu.6202@osu.edu (M.L.); niu.214@osu.edu (X.N.); yadav.94@osu.edu (K.K.Y.); lee.8757@osu.edu (C.L.); saif.2@osu.edu (L.J.S.); 2UC Davis Institute for Regenerative Cures, Department of Dermatology, School of Medicine, University of California Davis, Sacramento, CA 85817, USA

**Keywords:** porcine delta coronavirus, sparrow delta coronavirus, S protein, cross-species infection, turkey poults, chicken embryos, coronaviruses

## Abstract

Avian species often serve as transmission vectors and sources of recombination for viral infections due to their ability to travel vast distances and their gregarious behaviors. Recently a novel *deltacoronavirus* (DCoV) was identified in sparrows. Sparrow *deltacoronavirus* (SpDCoV), coupled with close contact between sparrows and swine carrying porcine deltacoronavirus (PDCoV) may facilitate recombination of DCoVs resulting in novel CoV variants. We hypothesized that the spike (S) protein or receptor-binding domain (RBD) from sparrow coronaviruses (SpCoVs) may enhance infection in poultry. We used recombinant chimeric viruses, which express S protein or the RBD of SpCoV (icPDCoV-S_HKU17_, and icPDCoV-RBD_ISU_) on the genomic backbone of an infectious clone of PDCoV (icPDCoV). Chimeric viruses were utilized to infect chicken derived DF-1 cells, turkey poults, and embryonated chicken eggs (ECEs) to examine permissiveness, viral replication kinetics, pathogenesis and pathology. We demonstrated that DF-1 cells in addition to the positive control LLC-PK1 cells are susceptible to SpCoV spike- and RBD- recombinant chimeric virus infections. However, the replication of chimeric viruses in DF-1 cells, but not LLC-PK1 cells, was inefficient. Inoculated 8-day-old turkey poults appeared resistant to icPDCoV-, icPDCoV-S_HKU17_- and icPDCoV-RBD_ISU_ virus infections. In 5-day-old ECEs, significant mortality was observed in PDCoV inoculated eggs with less in the spike chimeras, while in 11-day-old ECEs there was no evidence of viral replication, suggesting that PDCoV is better adapted to cross species infection and differentiated ECE cells are not susceptible to PDCoV infection. Collectively, we demonstrate that the SpCoV chimeric viruses are not more infectious in turkeys, nor ECEs than wild type PDCoV. Therefore, understanding the cell and host factors that contribute to resistance to PDCoV and avian-swine chimeric virus infections may aid in the design of novel antiviral therapies against DCoVs.

## 1. Introduction

Coronaviruses (CoVs) are enveloped viruses possessing the largest positive sense, single-stranded RNA genomes [1]. CoVs belong to the family *Coronaviridae* and the order *Nidovirales* [2]. According to the phylogenetic relationships and genomic structures, there are four genera of the *Orthocoronavirinae* subfamily: *Alphacoronavirus*, *Betacoronavirus*, *Gammacoronavirus* and *Deltacoronavirus*. Viruses from each coronavirus genus have been found in diverse host species, but only DCoVs in multiple mammalian and avian species [1]. Within their natural hosts, CoVs cause respiratory and intestinal infections, peritonitis and neurological infections with a range of symptoms from mild to lethal disease [3,4]. Continued zoonotic spillover of coronaviruses from bat and avian hosts, which appear to serve as both reservoir and recombination sources, necessitates understanding the complex mechanisms associated with cross-species transmission. The emergence of severe acute respiratory syndrome (SARS) CoV [5], Middle East respiratory syndrome (MERS) CoV [6], and SARS-CoV-2, the causative agent of the current global pandemic coronavirus disease 2019 (COVID-19) [7], underscores the importance of understanding CoV host switching. The genome arrangements of *delta-coronaviruses* (DCoVs), which have the smallest RNA genome among known CoVs (26 kb in length), are: open reading frame 1ab (ORF1a/1b), spike protein (S), envelope protein (E), membrane protein (M), nucleocapsid protein (N), and accessory proteins [8]. The S protein plays a major role in determination of cellular susceptibility of tissue and host [9,10].

S protein is homo-trimeric and consists of two subunit domains, S1 and S2 [11]. The S1 domain recognizes and binds the cell surface receptor, and S2 domain mediates membrane fusion [12]. The gene encoding the S protein is located within the region at which CoV genetic recombination occurs most frequently [13]. This high variation of the S protein is thought to be a primary factor leading to host and tissue tropism changes along with escape from viral neutralizing antibodies generated by the host [14].

While gamma-coronaviruses have been recognized as disease causing agents in avian species for many decades, DCoVs have been characterized only recently in avian species (2006) and even more recently in pigs in 2009 [15,16]. Porcine CoV HKU15, the prototype strain of porcine delta-coronavirus (PDCoV), is a novel enteropathogenic coronavirus, initially identified in pig fecal samples collected in Hong Kong in 2009 [15]. The contribution of PDCoV to significant clinical disease was unclear until early 2014, when it was detected in fatal cases of diarrhea in piglets in the United States (US) [17]. The most recent common ancestor of PDCoV and avian DCoVs is predicted at 523 years, but is likely to have occurred much sooner when accounting for recombination of the S gene [16]. Mutation and recombination enable CoVs to bind to new receptors and to enter host target cells [18], primarily through the S protein [19]. PDCoV has been identified globally including in the US [8], China [20], South Korea [21], Thailand [22], and Vietnam [23]. Birds are thought to serve as the natural viral host reservoir [24], and to evolve and disperse DCoVs [25]. Similarly, birds are considered the reservoir of multiple emerging pathogens such as avian influenza [26]. Birds play a role in circulation and outbreaks of viral pathogens such as the influenza virus [27], because of their ability to fly for a long distances and their gregarious behaviors. Therefore, birds can disperse emerging viruses not solely among themselves but also to humans and animals [18]. Cross-species spread frequently occurs in DCoVs due to recombination, which commonly includes the spike protein [18]. The PDCoV genome sequence is closely related to sparrow coronavirus (SpCov) HKU17 with more than 90% amino acid identity [15]. Sequence analysis suggests that PDCoV originated from recombination events between the DCoV SpCoV HKU17 and bulbul DCoV HKU11 (BuCoV HKU11), which was described by Lau et al. in 2009 [18]. The identification of quail DCoV UAE-HKU30 (QuaCoV UAE-HKU30) with high sequence identity to PDCoV and SpCoV HKU17, further supports the bird to swine transmission hypothesis [18]. Unexpectedly, full length genome sequences of sparrow CoV in fecal samples collected from wild birds in pig barns in the US showed higher identity with PDCoV and HKU17 than with other avian DCoVs [28], supporting the potential presence of an ecological bridge reservoir between PDCoV and SpDCoVs [29].

Interactions between wild birds, which carry DCoVs, and swine housed in hoop-style buildings, allowing regular exposure to wild birds, might be a source of viral recombination. These potential recombination events may help PDCoV adapt to domestic avian species, posing a risk of infection to poultry. Recently, the S receptor binding domain of PDCoV was shown to utilize a conserved motif within host aminopeptidase N (APN), allowing for binding and infection of cell cultures from diverse species, including humans and chickens [19]. Boley et al. demonstrated that PDCoV can infect 14-day-old chicks and 14-day-old turkey poults [25]. Jung et al. experimentally infected gnoto-biotic calves with PDCoV [30]. More recently it was shown that replacement of the S protein of PDCoV with the spike protein from SpDCoV resulted in asymptomatic infection of pigs with altered tissue tropism from the gastrointestinal tract to the respiratory tract [31]. The consequence of recombination of PDCoV with SpDCoV S in infection of poultry has not been reported. Therefore, this study was conducted to investigate the effects of SpDCoV S protein or RBD replacement in an infectious clone of PDCoV on viral susceptibility and replication in various cell lines, on pathogenesis and pathology in specific pathogen free (SPF) turkey poults, and on the infection of embryonated chicken eggs. 

## 2. Materials and Methods

### 2.1. Viruses

icPDCoV, icPDCoV-S_HKU17_ and icPDCoV-RBD_ISU_ were produced and characterized by Dr Q Wang and colleagues as described previously [31]. These viruses were validated for infectivity in gnotobiotic pigs as described [31].

### 2.2. Cells

DF-1—chicken embryo fibroblast cell line—(CRL-12203, American Type Culture Collection) (ATCC) and LLC-PK1—porcine kidney cell line—(CL-101, ATCC) were cultured in growth medium containing Dulbecco’s Modified Eagle’s Medium (DMEM) (Life Technologies, Carlsbad, CA, USA), supplemented with 10% heat-inactivated fetal bovine serum (FBS) (Life Technologies) and 0.1% penicillin-streptomycin (Thermo Fisher, Waltham, MA, USA), or Minimal Essential Media (MEM) (Life Technologies, Carlsbad, CA, USA), supplemented with 5% FBS, 1% anti-anti (Gibco), 1% non-essential amino acids (Thermo Fisher, Waltham MA, USA), and 1% HEPES (Thermo Fisher, Waltham MA, USA), respectively. Cell lines were incubated at 37 °C and 5% CO_2_ atmosphere. icPDCoV, icPDCoV-S_HKU17_, and icPDCoV-RBD_ISU_ viruses were propagated on LLC-PK1 cells in MEM, supplemented as described above plus 10 µg/mL trypsin (Thermo Fisher, Waltham MA, USA). Cells that had attained 80% confluence were infected for one hour prior to addition of primary growth media without FBS. Cells were allowed to grow for 48–60 h until optimal cytopathic effects were observed. LLC-PK1 cells were then frozen and thawed once, centrifuged, and the supernatants were aliquoted and stored as viral stocks at −80 °C until use. Viral titers were determined by cell culture immunofluorescence assay (CCIF), as previously described [32,33] or via median tissue culture infectious dose (TCID_50_). For CCIF, LLC-PK1 cells were grown in 96-well microtiter plates, and ten-fold serial dilutions of the viruses in MEM, supplemented with 1% anti-anti, 1% non-essential amino acids, 1% HEPES, and 10 µg/mL trypsin, were inoculated into each well. Eighteen hours after inoculation, cells were washed twice with phosphate-buffered saline (PBS) and fixed by incubation with 80% acetone for 10 min at room temperature (RT). Cells were stained and incubated with a mouse monoclonal antibody, SD55-197 against PDCoV N protein (NP) (www.medgenelabs.com accessed on 1 June 2021) at a concentration of 1:500 at 4 °C overnight, followed by incubation with 1:400 Alexa Fluor 488 conjugated goat anti-mouse IgG antibodies (Invitrogen, Carlsbad, CA, USA) for 1 h at 37 °C. Cell nuclei were visualized with 4′,6-diamidino-2-phenylindole (DAPI) (Thermo Fisher, Waltham, MA, USA), and virus-infected cells were visualized using an Olympus IX-70 fluorescent microscope (Olympus, Tokyo, Japan) and quantified.

### 2.3. Replication Kinetics in Various Cell Lines

The replication of icPDCoV, icPDCoV-S_HKU17_ and icPDCoV-RBD_ISU_ in DF-1 and LLC-PK1 cell lines was compared using a multiplicity of infection (MOI) of up to 10 and 0.01, respectively. Viruses were incubated with the indicated cells in triplicate wells of a 96-well plate or 24 well plate at 37 °C and 5% CO_2_ for 1 h, as previously described [31]. After removal of virus inocula, cells were washed twice with PBS. Cells were covered with serum-free media with 10 µg/mL trypsin (LLC-PK-1) or no trypsin (most DF-1 assays) (Gibco) and incubated at 37 °C and 5% CO_2_. Supernatants were harvested at 0, 6, 12, 24, 48, and 72 h and stored at −80 °C until use or entire wells were scraped at 6, 12, 24, 48, and 72 hpi. The replication kinetics were performed in triplicate. CCIF or TCID_50_ was used to titrate the viruses as described previously [32]. The results are presented as mean and standard deviation of fluorescent foci-forming units or TCID_50_ per mL. 

### 2.4. Cell-Associated and Cell-Free Virus RNA Loads in DF-1 Cell Line

The cell-associated and cell-free viral loads were assessed as genome equivalent of viral RNA in inoculated DF-1 cells collected during the experiment. DF-1 cells were cultured in growth medium as described earlier in a 96 well plate at 37 °C and 5% CO_2_. Cells that had attained 90% confluence were infected with icPDCoV-, icPDCoV-S_HKU17_-, and icPDCoV-RBD_ISU_ viruses at an MOI of 1. Viruses were incubated with the indicated cells in duplicate wells at 37 °C and 5% CO_2_ for 1 h, after which inocula were removed, and cells were washed twice with PBS. Cells were then covered with serum-free media with 10 µg/mL trypsin (Gibco) and incubated at 37 °C and 5% CO_2_. Cell-associated and cell-free virus were harvested at 0-, 6-, 12-, and 24-h post infection (hpi) and stored at −80 °C until assayed (Figure 1). Before extraction of the cell-associated fraction, the cells in the monolayer were washed twice with PBS then dissolved in 200 µL of lysis buffer containing RXV and RNA carrier from RNA extraction kit and scraped from the plate. Cell-associated and cell-free viral loads were analyzed by quantification of viral RNA using a TaqMan-probe-based RT-qPCR with PDCoV-M gene-specific primers. The percentage of release was calculated using the following ratio:$$\mathrm{Release}\;\mathrm{ratio}=\frac{\mathrm{cell-free}}{\left(\mathrm{cell-associated}+\mathrm{cell-free}\right)}\times 100$$

### 2.5. Experimental Inoculation Turkey Poults

In vivo experimentation was undertaken within the guidelines of the Ohio State University. Protocols and chicken sampling procedures were approved by the Institutional Animal Care and Use Committee (IACUC). 

The pathogenesis and pathology of the infectious clone PDCoV and recombinant viruses were evaluated in SPF turkey poults (Meleagris gallopavo). A total of 56 seven-day-old SPF turkey poults were obtained from the Ohio Agricultural Research and Development Center of The Ohio State University (Wooster, Ohio, USA) flock (OARDC flock), which has no history of exposure to swine or symptoms related to PDCoV, TGEV, or PEDV. Poults were floor housed in temperature-controlled biosafety level 2 (BSL-2) rooms for the period of the experiment with sustained artificial light, floor covered with wood litter shavings and birds were continuously provided access to food and water. Birds appeared healthy and showed no diarrhea or other clinical signs during the acclimation period. Turkey poults were randomly assigned to one of the four infection groups, which were housed in separate rooms: (1) icPDCoV-infected (*n* = 10); (2) icPDCoV-S_HKU17_-infected (*n* = 10); icPDCoV-RBD_ISU_-infected (*n* = 10); or MEM-infected (control) (*n* = 10). Birds were inoculated through the choanal cleft with 4.4 × 10^5^ FFU/poult of virus in a volume of 195 μL. Ten control turkey poults were also inoculated with the same amount of MEM. Four uninfected birds per group, at 1-day post inoculation (DPI), were randomly assigned to serve as sentinels and allowed to commingle with each infected or control group (Figure 2).

### 2.6. Clinical Swabs and Tissue Collection

To evaluate viral shedding, tracheal and cloacal swabs were collected from challenged and control poults daily. Fecal consistency produced during swabbing and via room observation was monitored and scored as follows: 0, solid (No diarrhea); 1, pasty (Likely normal); 2, semiliquid (Some potential diarrhea); and 3, liquid (Diarrhea) as previously published [25]. Fecal scores >2 were considered as diarrhea. Tracheal and fecal samples were diluted in 1 mL MEM and centrifuged at 1832× *g* at 4 °C for 20 min, and the supernatants were collected for viral RNA isolation. Five poults from each group were euthanized and their duodenum, jejunum, ileum, and lung were collected at 3- and 5- DPI and fixed with 4% paraformaldehyde during the bird necropsy. All sentinel birds were euthanized at 3-day post commingling. Paraformaldehyde-fixed tissues were preserved for immunohistochemical examination to detect lesions.

### 2.7. Inoculation of Chicken Embryos with the Chimeric Viruses

A total number of 29 11-day-old SPF ECEs and 55 5-day-old SPF ECEs from the OARDC flock were divided into 4 or 3 groups, respectively. Viruses icPDCoV, icPDCoV-S_HKU17_, icPDCoV-RBD_ISU_ were injected into the 29 11-day-old SPF ECEs using a titer of 6 log_10_ fluorescent focus units (FFU)/egg, in 200 µL. Five ECEs for each virus were inoculated with viable icPDCoV, icPDCoV-S_HKU17_, or icPDCoV-RBD_ISU_. As a background control, three ECEs for each virus were similarly inoculated with killed (heat inactivated at 60 °C for 20 min) viruses. The ECEs of the mock group were inoculated with 200 µL MEM. For the 5-day-old SPF ECEs, icPDCoV, icPDCoV-SHKU17, or icPDCoV-RBDISU were serially diluted from neat to up to 10^−6^ in MEM and used to inoculate 5 eggs per infectious dose, 100 µL/egg. In brief, the embryos were inoculated via the allantoic cavity route and incubated at 37.5 °C, 55 ± 2% relative humidity. The viability of embryos was examined by using an egg candler. An ECE was harvested by putting at 4 °C for 2–6 h when it was found dead. The remaining 11-day-old and 5-day-old SPF ECEs were harvested at 3 DPI and 5 DPI, respectively. The eggshells above the air sac were disinfected with 70% ethanol, and the allantoic fluids were collected and centrifuged at 4122× *g* at 4 °C for 20 min to remove the cell debris. For the trial using 11-day-old ECEs, after removing heads, wings, and legs, embryos were dissected into two portions, and the whole thoracic and abdominal tissues in each cavity were harvested separately and homogenized using an electric homogenizer (OMNI GLH International/USA) for 1 min in equal amounts of MEM (1 g of tissue:1 mL of MEM). The allantoic fluids and tissues were analyzed by quantification of viral RNA using a TaqMan RT-qPCR with PDCoV-M gene-specific primers as reported previously [34]. The allantoic fluid samples of the 5-day-old ECEs (at no dilution and 1:10 dilution) were tested for infectious virus in LLC-PK1 cells using 96-well plates.

### 2.8. RNA Isolation and RT-qPCR Analysis

Total viral RNA was extracted from 150 µL of swab supernatants using TRIzol reagent (Life Technologies, Carlsbad CA, USA) according to the manufacturer’s instructions and eluted in 40 µL RNase free water. We determined the titers of viral RNA shed in swabs by carrying out one step TaqMan-probe-based RT-qPCR as described previously [35] targeting the PDCoV-specific M gene with the primers (forward 5′-ATC GAC CAC ATG GCT CCA A-3′ and reverse 5′-CAG CTC TTG CCC ATG TAG CTT-3′), which were designed based on the sequence of PDCoV strain US, Illinois121/2014 (GenBank accession no. KJ481931) and the probe (5′-/56-FAM/CAC ACC AGT/ZEN/CGT TAA GCA TGG CAA GCT/3IABkFQ/3′). The reaction system was set up with 12.5 µL nuclease free water, 5 µL of 4X TaqMan (Applied Biosystems, Waltham MA, USA), 0.5 µL of F primer, R primer, and probe, and 1 µL of template RNA. The amplified fragment was 541-bp with the following thermal cycling profile: 50 °C for 5 min and 95 °C for 20 s, followed by 45 cycles of 95 °C for 3 s and 55 °C for 30 s. The detection limit of the RT-qPCR was 1.47 × 10^5^ genomic equivalents (GEs)/mL, which corresponded to 5.17 log_10_ GE/mL of PDCoV.

### 2.9. Immunohistochemistry (IHC) and Immunofluorescence (IF) Staining of Tissues from Turkey Poults

At necropsy, tissues including small intestines, duodenum to ileum, and lung were collected from euthanized poults and then fixed in 4% paraformaldehyde for 24 h at room temperature. After trimming, processing, embedding, and sectioning (4.5 μm), tissues were first incubated at 60 °C for 30 min and deparaffinized using xylenes followed by rehydrating through graded alcohol and deionized (DI) water [36,37]. 

For IHC analysis, dewaxed sections were treated with pronase reagent (GeneTex), for 18 min at RT for antigen retrieval followed by peroxide and power block (BioGenex, Fremont, CA, USA) to quench endogenous peroxide and non-specific binding for 10 and 30 min at RT, respectively. Sections were then incubated with a primary antibody as above at 4 °C overnight, followed by a non-biotin polymerized horseradish peroxidase system (BioGenex Laboratories, Fremont, CA, USA) as described previously [31]. PBS or deionized water was used for rinsing cells between incubations. Slides were imaged using an Olympus IX-70 microscope (Olympus, Tokyo, Japan).

For IF staining, DF-1 and LLC-PK1 cells were grown on sterilized glass coated with poly-L-lysine (Fisher Scientific, Waltham MA, USA) in a 6-well plate and inoculated at an MOI of 0.01. Cell lines were subsequently fixed in 4% paraformaldehyde at 18 hpi and 12 hpi, respectively, followed by epitope unmasking using 0.05% tween phosphate buffer solution (TPBS) for 10 min at room temperature (RT). Non-specific binding sites on cells were blocked with 0.1% power block universal blocking reagent X-10 (BioGenex, Fremont, CA, USA) in distilled water for 30 min at RT. Cells were then incubated with mouse anti-N monoclonal antibody (mAb) SD55-197 diluted 1:500 for 2 h at RT, followed by incubation with goat anti-mouse IgG antibody conjugated with Alexa Fluor 488 at a dilution of 1:400 (Invitrogen, Carlsbad, CA, USA) for 90 min at RT and nuclei were visualized using DAPI (Thermo Fisher, Waltham, MA, USA) for 3 min at RT. Phosphate buffered saline (PBS) was used for rinsing cells between incubations and PBS with 0.2% bovine serum albumin (BSA) was used for antibody dilution. Cells were observed using confocal microscopy (Leica, Wetzlar, Germany).

### 2.10. Statistical Analysis

Statistical analysis was performed with two-way ANOVA and Bonferroni post hoc test or one-way ANOVA with Tukey’s post hoc test using GraphPad Prism 5. Differences in means between groups were considered significant when the *p* value was less than 0.05. Results are expressed as mean ± SD of the means.

## 3. Results

### 3.1. icPDCoV and Recombinant Viruses in In Vitro Cultures

To determine if icPDCoV, icPDCoV-S_HKU17_, and icPDCoV-RBD_ISU_ viruses would infect chicken and swine cell lines, mock and infected chicken DF-1 or swine LLC-PK1 cells were stained for viral antigen. Notably, we demonstrated that DF-1 and confirmed that the LLC-PK1 cells are susceptible to infection by the icPDCoV and chimeric viruses. Viral NP localized specifically at cytoplasmic regions in DF-1 cells (Figure 3C,E,G) and in LLC-PK1 cells (Figure 3D,F,H) consistent with previous results for LLC-PK1 [31], while mock-inoculated cells had no NP staining (Figure 3A,B). Taken together, these results indicate that DF-1 and LLC-PK1 cells are susceptible to icPDCoV, icPDCoV-S_HKU17_, and icPDCoV-RBD_ISU_ viral infections.

### 3.2. Replication Kinetics of Viruses in Chicken and Swine Cells

To examine whether viral replication kinetics differed among the viruses, we assessed and compared viral replication kinetics in DF-1 and LLC-PK1 cells across multiple, matched time-points. We observed that icPDCoV virus produced significantly higher viral titers in supernatant fractions harvested from LLC-PK1 cells than recombinant viruses when infected at an MOI of 0.01 (Figure 4A). Attempts to infect DF1 cells at MOIs of 0.1, 1, 5, and 10 resulted in no detectable infectious particles released at any tested timepoints as assessed with CCIF (Data not shown), indicating that the replication of chimeric viruses in DF-1 cells was below the level of CCIF detection. Repeating the assay utilizing TCID_50_ showed a low infectious titer in which icPDCoV produced minimal replication while icPDCoV-S_HKU17_, and icPDCoV-RBD_ISU_ were unable to produce additional infectious particles (Figure 4A). A starting infectious dose of 10 MOI yielded infectious titers for each test virus which gradually declined over time, suggesting that virus particles attached to cells but failed to replicate (Figure 4B). Distinct differences in observed cytopathic effect between each virus were noted with icPDCoV producing larger cellular syncytia than icPDCoV-S_HKU17_ and icPDCoV-RBD_ISU_ which retained distinct cellular borders (Figure 4C,E). Altogether, the results obtained showed that DF-1 cells are likely less permissive than LLC-PK1 for infection with icPDCoV-S_HKU17_, and icPDCoV-RBD_ISU_ with potential differences in S-mediated syncytia formation. 

### 3.3. In Vitro Assessment of Viral RNA Loads in Inoculated DF-1 Cells

Viral RNA loads at longitudinal time points through 24 h post inoculation were measured by RT-qPCR to assess in vitro viral RNA levels. We compared the viral RNA loads from the cell-associated fraction to the viral RNA loads in the same inoculated DF-1 cells of the cell-free fraction. Predictably, we found significant differences in the viral RNA loads between cell-associated fraction (*n* = 48) and cell-free fraction (*n* = 48) (*p* = 0.05) (Figure 5A–D). Consistent with the findings from replication kinetics (Figure 4B). (Cell-associated virus appeared to remain static over time (Figure 6A) while cell free viral RNA increased from the 6–12 h timepoints but remained static from 12–24 h (Figure 6B). We further examined the total release ratios of the DF-1 inoculated with icPDCoV-, icPDCoV-S_HKU17_- or icPDCoV-RBD_ISU_ viruses to determine the presence of cell-associated virus in the inoculated DF-1 cells. Indeed, we found no differences in viral release between chimeric viruses (Figure 7A–C). These results suggest that the reduction in infectious virus observed at MOIs lower than 40 was not attributable to viral release, suggesting that there might be other mechanistic pathways contributing to the decrease in the infectious virus in cell-free fractions; however, in this study we did not determine these mechanisms.

### 3.4. Clinical Manifestations in SPF Turkey Poults and RT-qPCR of Viral RNA in Tracheal and Fecal Samples

Fecal consistency scores at 1, 3, and 5 PDI were significantly higher in groups treated with icPDCoV-S_HKU17_ virus or icPDCoV-RBD_ISU_ virus compared to control group (Figure 8A). Moreover, uninfected sentinel birds that commingled with icPDCoV-S_HKU17_- or icPDCoV-RBD_ISU_- infected group had higher fecal score at 2 DPI compared to control infection (Figure 8B). Consistent with the results from the fecal consistency score, at 1 DPI, the groups infected with icPDCoV-S_HKU17_ or icPDCoV-RBD_ISU_ virus were more lethargic (movement less frequently) and poorly feeding more than the icPDCoV infected group, suggesting that the spike protein could play a role in virus infection. However, cloacal and tracheal swabs from experimentally infected turkey poults were negative for viral RNA throughout the study. At the cutoff value of 5.17 log_10_ GE/mL, no samples tested higher than control samples. The results indicate that both cloacal and tracheal swabs were negative for all viruses tested although mild clinical signs were noted.

### 3.5. Gross and Immunobiological Examination of PDCoV Pathology and NP Immunoreactivity in Inoculated Poults

Although turkey poults from the virus-inoculated groups showed mild signs of intestinal congestion, and intestinal gas accumulation, and those from icPDCoV-S_HKU17_ group showed a fecal score of 2, briefly, after standard IHC criteria and mAb incubation, IHC staining showed no cellular signal for positive PDCoV NP in the lung or small intestinal tissue sections that were included in this study (Figure 9), which is consistent with our observed in vitro results, that DF-1 cells are not permissible to the virus infections at infectious MOI of 0.1, 1, 5, and 10. These data demonstrate that while icPDCoV and chimeric viruses seemed to cause mild gross and histopathological complications in turkey poults, we were unable to detect significant viral replication.

### 3.6. Absence of Viral RNA Titer Increases in 11-Day Old ECEs Indicate Lack of Viral Replication

We performed an RT-qPCR assay using M gene specific primers for the detection of viral RNAs from total RNA extracted from allantoic fluid (*n* = 29), thoracic- (*n* = 29), and abdominal- (*n* = 29) tissues. A total of 87 samples from three groups were tested. ECEs were inoculated by the allantoic cavity with heat-inactivated (*n* = 3 for each virus) or viable (*n* = 5 for each virus) icPDCoV-, icPDCoV-S_HKU17_-, icPDCoV-RBD_ISU_ viruses, or mock-inoculated (*n* = 5). Our data indicated that the icPDCoV-, icPDCoV-S_HKU17_- or icPDCoV-RBD_ISU_ RNAs were detected in allantoic fluids at 3 DPI, and the viral RNA titers were significantly higher in allantoic fluids than in the control group (Figure 10A). However, viral RNA from the heat inactivated controls remained at the same level as the live virus, suggesting that no significant replication occurred. Similarly, levels of the viral RNAs quantitated from thoracic- and abdominal tissues were not statistically different between infected and mock-inoculated groups and also did not deviate from the heat inactivated viral RNA titers (Figure 10B,C), suggesting that icPDCoV and chimeric viruses did not replicate in the 11-day-old ECEs. Moreover, no gross lesions or pathological changes or embryo deaths were observed or recorded in the present study. These findings are consistent with a recent report of PDCoV being unable to propagate in 8-day-old ECEs [38].

### 3.7. Differential Virus Replication in 5-Day-Old ECEs

After inoculating 5-day-old ECEs with serially diluted icPDCoV-, icPDCoV-S_HKU17_- or icPDCoV-RBD_ISU_ (100 µL/ECE), we observed 100% (5/5) ECE death in icPDCoV-inoculated ECEs at the 10^−2^ through 10^−5^ dilutions, whereas icPDCoV-RBD_ISU_ had limited ECE death rates of 50% (3/6) at 10^−2^ dilution and 40% (2/5) at the 10^−4^ dilution only, and icPDCoV-S_HKU17_ was unable to produce ECE death at any infectious titer by 5 DPI (Table 1). The allantoic fluid samples (at no dilution and 1:10 dilution) were tested for infectious virus in LLC-PK1 cells. Selected samples were tested by DCoV-specific RT-PCR. We found that the infectivity assay was as sensitive as the RT-PCR. These results indicate that wild type PDCoV was the most capable at entering and replicating within 5-day-old ECEs, suggesting that the RBD or S from sparrow CoVs was less able to support PDCoV entry into chicken cells or a potential synergism within and between S protein and other viral factors is needed for optimal virus replication. 

## 4. Discussion

Emerging infectious diseases are frequently diagnosed in poultry species. Thes3 emerging infectious diseases can arise due to the cross-species transmission of infectious agents, and these crossover infections may pose a serious threat to public and animal health. For instance, the CoVs, SARS-CoV in 2002–2003, MERS-CoV in 2012, and SARS-CoV-2 since 2019 have recently originated in reservoir animal hosts (likely in bats) and been transmitted to humans [19]. Similarly, highly pathogenic avian influenza (HPAI) A (H5N1) viruses that originated in poultry, have raised concerns for transmission to humans and have spread worldwide [39,40]. Additionally, a novel avian-origin influenza A (H7N9) emerged in 2013 in southeast China, spread among humans, and caused deaths in one third of the patients due to severe lower respiratory infection [41,42].

The S protein of coronaviruses plays a critical role in the recognition of host cellular receptors and mediation of membrane fusion [31]. Genetic recombination and mutations occur most frequently in the S protein [13]. The emergence of multiple new variants of concern of SARS-CoV-2 are attributed to multiple mutations in the S protein. Additionally, some of the S protein mutations can lead to enhanced viral infectivity or transmissibility, as evidenced by the D614G mutation in the S protein of SARS-CoV-2 [43]. Mutations within CoVs arise in three ways: (1) Intrinsically, due to the errors that occur during viral replication [44]; (2) genomic variability, which arises when two viral lineages infect the same host [45]; and (3) due to host RNA-editing systems, which are considered part of the natural viral defense mechanism [46]. It has been stated that the majority of mutations are neutral [47]. However, some mutations may increase the virulence and/or transmission and others are deleterious to the virus [43,48]. Advantageous and neutral mutations have higher frequencies [47,48].

In the present study, we used chimeric viruses that express spike (S) protein of SpCoV HKU17 or the RBD of SpDCoV ISU73347 on the genomic backbone of an infectious cDNA clone of virulent PDCoV OH-FD22 strain (icPDCoV). The first fundamental step in viral infection is the receptor interaction. We demonstrate that chicken DF-1 and confirm that swine LLC-PK1 cells—derived from chickens and pigs, respectively—are susceptible to icPDCoV-, icPDCoV-S_HKU17_- and icPDCoV-RBD_ISU_ virus infections, suggesting that chimeric viruses exhibit a broad species cell tropism, infecting cells derived from both chickens and swine. Our previous data demonstrated that PDCoV employs aminopeptidase N (APN) for cell entry, and PDCoV can infect cell lines derived from chickens and humans [19]. Therefore, chimeric viruses in the present study might employ APN as a receptor to enter DF-1 cells. However, investigators have recently found that upon infection of APN-knockout ST cells, the replication of icPDCoV-S_HKU17_ virus remained at the same level as its replication in wild type ST cells, but the replication of icPDCoV-RBD_ISU_ virus increased significantly compared to the wild type ST cells [31]. Coupled with the ability of DCoVs to infect hosts from various species such as birds and mammals, this suggests that chimeric viruses might employ a broad range of receptors. Furthermore, it has been suggested that the use of APN or angiotensin-converting enzyme 2 (ACE2) as a receptor was independently selected, according to CoV evolution [19,49]. Although chimeric viruses replicated to significantly higher titers in DF-1 cells, their replication was not optimal since they only began to produce virus measurable via FFU at an MOI of 40, whereas chimeric viruses and icPDCoV showed replication in LLC-PK1 cell line at an MOI of 0.01. These findings indicate that one or more mechanistic pathways and cellular factors play an essential role in enabling the permissiveness since the cycle of viral replication is a complex process. Here, we suggest that two mechanistic pathways could have adverse consequences on permissiveness: first, we hypothesize that a strong upregulation of interferon responses, in particular type I interferons, may play a role in preventing new virus progeny infection if present within the supernatant applied to naïve cells in the FFU assay. A direct correlation between interferon responses and viral load has recently been found [50]. Second, we hypothesize that DF-1 cells may not be permissible due to reasons related to the receptors such as APN or other specific receptors. For instance, several studies demonstrated that SARS-CoV could not replicate in Madin-Darby Canine Kidney (MDCK) cells [51,52]. However, the replication of SARS-CoV was promoted and observed in engineered cells that express transmembrane serine protease 2 (TMPRSS2) that promotes change of the spike to allow cell entry [53]. The mechanisms for in vitro inhibition of the replication of chimeric and icPDCoV viruses need to be further investigated in the future.

In the inoculated turkey poults, immunohistochemistry analyses showed no signal against the N protein of PDCoV in small intestinal and lung tissues. The IHC data are consistent with the findings [31] that pigs showed no diarrhea and clinical signs, no villous atrophy, no intestinal lesions, and showed no viral antigens in small intestinal tissues after being orally/oro-nasally inoculated with the icPDCoV-S_HKU17_-, and icPDCoV-RBD_ISU_ virus. Consistent with our findings, Liang et al. [34] recently observed that PDCoV inoculated 4-day-old SPF chickens exhibited no clinical signs, and their body temperatures were the same as control group. More recently an in vitro study found that PDCoV replication was inhibited by bile acids chenodeoxycholic acid (CDCA) and lithocholic acid (LCA) due to their antiviral activity by stimulating interferons, λ3 and ISG15 [54]. Furthermore, none of the cloacal and tracheal swabs of inoculated birds and sentinel birds were clearly positive using RT-qPCR. Our results are further consistent with a previous report in which piglets did not shed viral RNA in feces following icPDCoV-S_HKU17_- and icPDCoV-RBD_ISU_ virus infections, nor did commingled pigs, indicating no pig-to-pig transmission [31]. The derivation of a single homogenous icPDCoV sequence based upon virus passaged eight times in tissue culture may have led to genomic mutations rendering the icPDCoV virus less able to infect and cause pathology in vivo than the virus inoculum derived from intestinal contents of gnotobiotic pigs. Additional studies comparing infectivity of icPDCoV and intestinal content-derived PDCoV along with virus sequence comparisons are needed to determine if specific mutations are responsible for a reduction of infectivity observed for the icPDCoV derived virus in inoculated turkey poults.

Differences between our studies and the Boley et al. results may be attributed to the use of the icPDCoV-derived clone. As mentioned above, the icPDCoV-derived virus may produce a more homogeneous starting virus (potentially clonal) versus the large intestinal contents utilized in the Boley et al. study. The viral material in that study was derived from LLC-PK1 tissue culture passage 20 virus which was also passaged through gnotobiotic pigs potentially allowing mutations and quasi-species that might have enhanced pathology in poultry. Additional differences exist between previous reported diarrhea, cloacal and tracheal viral shedding and enteric positive cells, and our present observations [25]. There are several hypotheses that account for these potential differences: First, the poults included in the present study were younger than the age of the poults in the previous study [25]. Age may play an important role in infectivity. Age is a noted factor in CoV pathology including with COVID-19 [55,56]. From this and other published studies there appears to be a window of 2–6 days in ovo, followed by days 11–24 post hatch in which poults are more susceptible to detectable infection [25,38,57]. Further infection studies with varied ages are needed to assess the ages in which chicks and poults are most susceptible to PDCoV infection. One potential age-related complication could be the quantity and localization of APN or the coreceptor. For example, the expression levels of α2,3SA-gal and α2,6SA-gal receptors in respiratory and intestinal tissues important for avian influenza infection were related to the age and species of poultry [58]. Additionally, we cannot rule out potential effects of maternal immunity in our study. Although the turkey poults in our study were from SPF flocks with no known prior exposure to PDCoV, they have not been tested to confirm they are seronegative for PDCoV. Thus, passively acquired immunity in hatching birds, which continues in the first 10 days of the life [59], could influence the susceptibility and permissiveness of these poults to viral infections. There is still a need to identify and determine of the tissue tropism of PDCoV in poultry. There is also a need to characterize the specific receptors and cellular proteases utilized by PDCoV and SpCoV. In our previous findings, we reported that PDCoV utilizes APN as an entry receptor because, in APN knockout cell lines, the susceptibility was drastically decreased and expression of APN in non-permissive cells allowed infectivity [19]. Recently, Niu et al. have found that, in APN-knockout ST cells inoculated with sparrow/swine chimeric viruses, the percentage of infectivity increased for icPDCoV-RBD_ISU_; however, the infectivity was the same as in wildtype ST cells for icPDCoV-SHKU17 infection [31]. Furthermore, in the same study, they found limited replication of recombinant chimeric viruses in the respiratory tract of inoculated piglets, whereas the viruses lost their tropism for the pig intestine.

Liang et al. reported that PDCoV could be propagated in 11-day-old ECEs, but also indicated that the levels of viral RNA were low, linking the reason behind their observation to incomplete viral adaptation [34]. For this study we initially hypothesized that sparrow CoV spike proteins may bestow a binding enhancement in avian species, therefore we attempted to infect 11-day-old ECEs. We were unable to detect significant replication in 11-day-old ECEs. 11-day-old ECEs inoculated with either heat-inactivated or viable icPDCoV-, icPDCoV-S_HKU17_-, icPDCoV-RBD_ISU_ viruses, or mock via the allantoic cavity produced no embryo death or increased viral RNA loads in ECEs (live vs. inactivated viruses), thus no evidence of viral replication was detected. These findings are consistent with the recent demonstration that > 8-day-old ECEs were not susceptible to PDCoV, suggesting that once the cells differentiate and become mature in ovo, they are not susceptible to PDCoV infection [38]. Upon repeating infectivity experiments, we were able to see embryo death in 5-day-old ECEs that were inoculated with icPDCoV, and to a lesser extent with icPDCoV-RBD_ISU_ but not icPDCoV-S_HKU17_. Further research is needed to determine whether replacement of the full S protein from ISU SpCoV would reduce replication levels to that of HKU17 SpCoV spike or whether S RBD insertions are simply more viable than full S replacements. Rather than sparrow CoV-derived RBD or spike protein bestowing enhanced binding of PDCoV to ECEs, this data indicates that an S-dependent adaptation in PDCoV has allowed for more efficient cross-species spread than ancestral sparrow S proteins. 

## 5. Conclusions

Our study presents the first qualitative and quantitative data for in vitro replication kinetics in chicken DF-1 cells, and ECEs of chimeric icPDCoV-S_HKU17_-, icPDCoV-RBD_ISU_ viruses. Our results indicated that the infectious clone-derived wildtype or chimeric viruses do not replicate in 11-day-old ECEs but icPDCoV and icPDCoV-RBD_ISU_ do replicate in 5-day-old ECEs with icPDCoV-S_HKU17_ unable to replicate efficiently even at the highest starting inoculum. Although the viruses replicated in chicken DF-1 cells, they required high starting MOIs and failed to maintain efficient replication. The findings of this study are important to better understand cross-species threat of *delta-coronaviruses* based on icPDCoV and the chimeric viruses bearing swine vs. avian RBD or S, respectively. Future work should be directed toward understanding genetic differences between infectious clone-derived PDCoV and chimeras versus porcine intestinal content-derived virus, along with the mechanistic pathways and cell and host factors that decrease infection in DF-1 cells and restrict infection of differentiated ECEs by the chimeric viruses. In addition, understanding the receptors utilized by chimeric viruses for cellular entry will be an important next step. This could aid in the design of novel antiviral protection against delta-CoVs. 

## Figures and Tables

**Figure 1 viruses-14-01225-f001:**
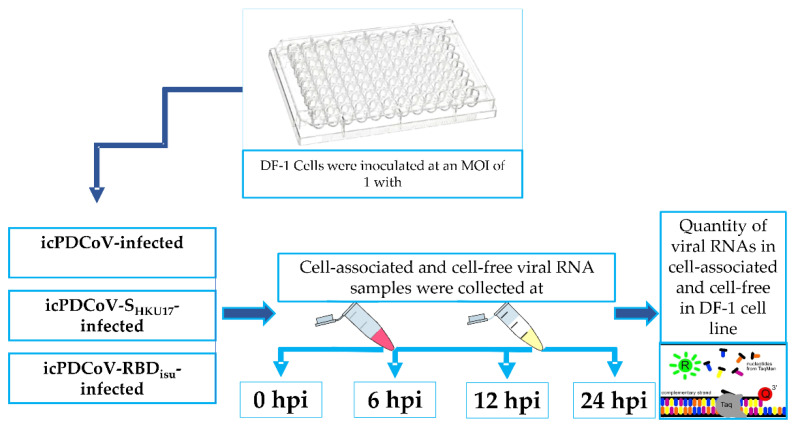
Schematic diagram of the DF-1 cells showing treatment groups and time points at which cell-associated and cell-free were collected for viral RNA titration.

**Figure 2 viruses-14-01225-f002:**
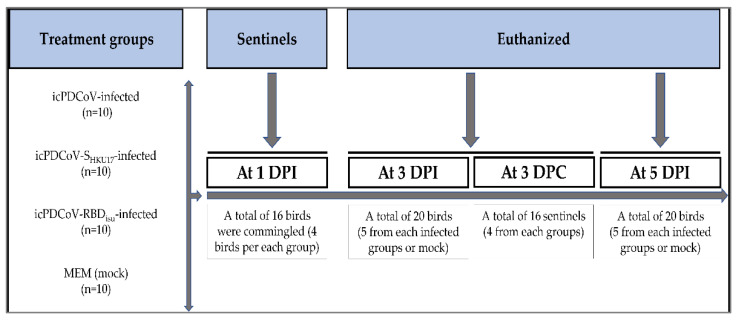
Schematic diagram of the experimental design showing treatment groups and time points at which the sentinels were commingled, and the birds were euthanized. Turkey poults at 8 days of age were infected and euthanized at 11 and 13 days of age. Sentinel birds at 10 days of age accordingly commingled with each infected group and euthanized at 12 days of age. Turkey poults inoculated with 4.4 × 10^5^ FFU/poult of icPDCoV-(passage 8), icPDCoV-S_HKU17_-(passage 3), and icPDCoV-RBD_ISU_ (passage 3), viruses, or MEM in a volume of 200 µL. DPC, day post commingling.

**Figure 3 viruses-14-01225-f003:**
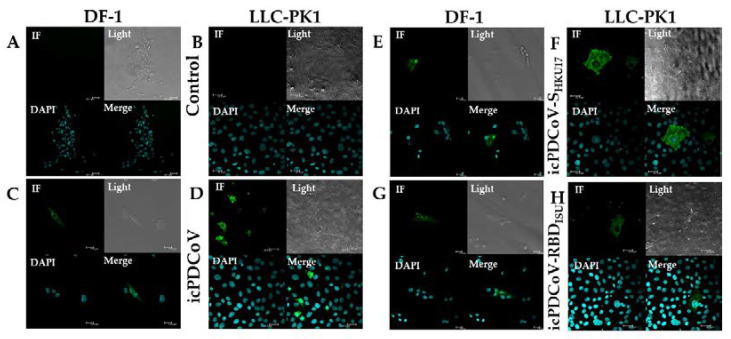
Indirect immunofluorescence imaging of PDCoV N antigen in DF-1 and LLC-PK1 cell lines. Cell lines were inoculated with mock inoculated (**A**,**B**), or inoculated with icPDCoV (**C**,**D**), icPDCoV-S_HKU17_ (**E**,**F**), and icPDCoV-RBD_ISU_ (**G**,**H**) viruses at an MOI of 0.01. DF-1 and LLC-PK1 cells were fixed at 18 hpi or 12 hpi and stained using a mouse monoclonal, SD55-197 against PDCoV N protein, respectively. Representative images are shown (labeling for PDCoV N antigen in the infected cells (in green) and nuclei (in blue)). Scale bara of the zoom: 1.5 are 22.5.

**Figure 4 viruses-14-01225-f004:**
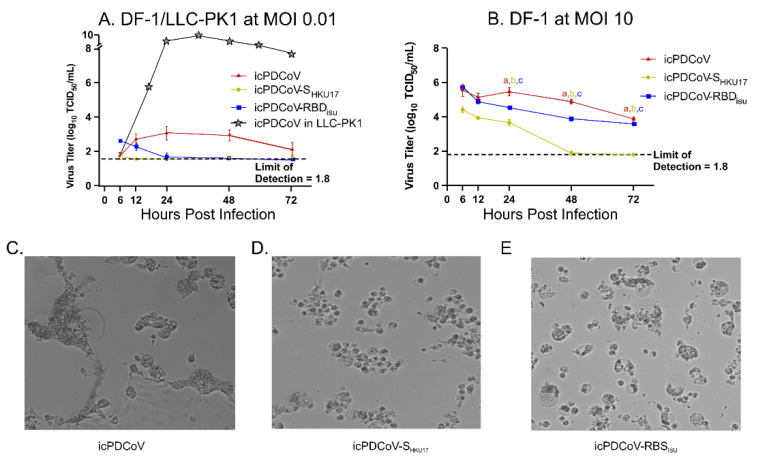
Replication kinetics of icPDCoV, icPDCoV-S_HKU17_, and icPDCoV-RBD_ISU_ viruses. (**A**) LLC-PK1 and DF-1 cell lines were inoculated at an MOI of 0.01 (4A) and DF-1 cells at an MOI of 10 (4B), respectively. LLC-PK1 kinetics at 0.01 MOI was reported previously by Niu et al. [31]. The data for icPDCoV were included in 4A as a TCID_50_ kinetics reference. After 1 h of incubation at 37 °C, cells were washed and overlaid with MEM supplemented with 10 µg/mL trypsin (LLC-PK-1) or no trypsin (DF-1) due to cell sensitivity. The kinetics of viral replication were evaluated in supernatants collected at denoted timepoints post infection, and the viral titers were determined by TCID_50_ (**A**,**B**). Different lower-case letters (a, b, c) indicate significant differences between each group (*n* = 3 biologically independent samples) at each time point. All data points are mean ± SD. Two-way ANOVA with Bonferroni post hoc test, *p* ≤ 0.05. ns, not statistically significant. (**C**–**E**) Light microscopy of DF-1 cells infected with specified virus at 72 h post infection with an MOI of 0.01.

**Figure 5 viruses-14-01225-f005:**
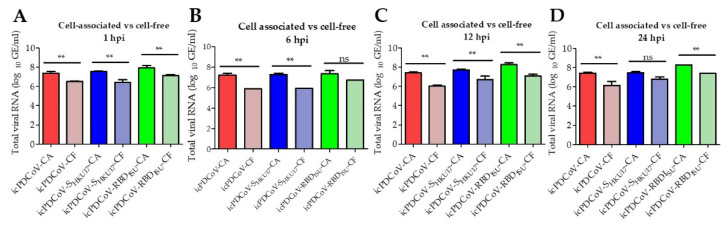
Comparison of viral RNA loads in inoculated DF-1 cells. The DF-1 cells were inoculated with icPDCoV, icPDCoV-S_HKU17_, and icPDCoV-RBD_ISU_ viruses at an MOI of 1. Data shown are levels of virus quantified in DF-1 from virus associated with DF-1 compared with the same inoculated DF-1 cells in cell-free fraction at (**A**) 0 hpi, (**B**) 6 hpi and (**C**) 12 hpi, and (**D**) 24 hpi. The cutoff value was 5.17 log10 GE/mL. hpi, hour post infection; GE, genomic equivalent; CA, cell-associated; CF, cell-free. One-way ANOVA with Tukey’s multiple comparisons post hoc test, ** *p* ≤ 0.05, ns; not statistically significant.

**Figure 6 viruses-14-01225-f006:**
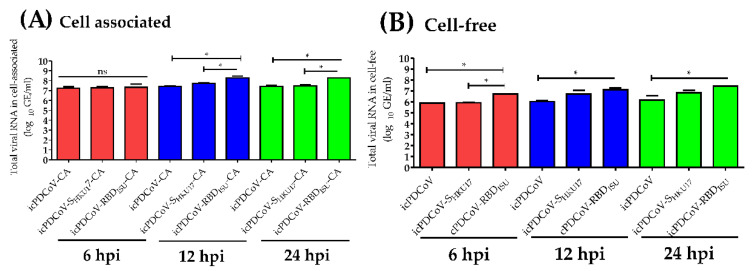
Comparison of viral RNA loads in virus associated with DF-1 and cell-free supernatant. The DF-1 cells were inoculated with icPDCoV, icPDCoV-S_HKU17_, and icPDCoV-RBD_ISU_ viruses at an MOI of 1. Data shown are levels of virus quantified in DF-1 from virus associated with DF-1 compared with the same inoculated DF-1 cells in cell-free fraction at 6 hpi, 12 hpi, and 24 hpi. hpi, hour post infection; GE, genomic equivalent; CA, cell-associated; CF, cell-free. One-way ANOVA with Tukey’s multiple comparisons post hoc test, * *p* ≤ 0.05, ns; not statistically significant.

**Figure 7 viruses-14-01225-f007:**
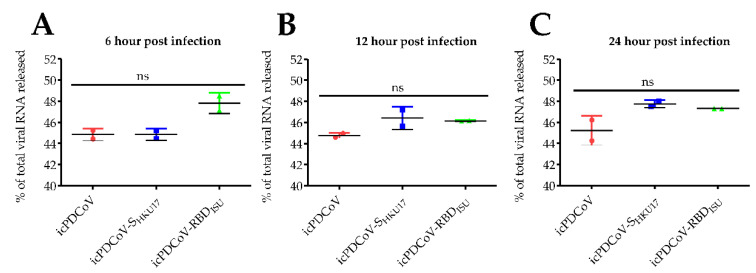
Released ratios of viral RNAs in inoculated DF-1 cells. The DF-1 cells were inoculated with icPDCoV, icPDCoV-S_HKU17_, and icPDCoV-RBD_ISU_ viruses at an MOI of 1. Data shown are levels of virus quantified in DF-1 from virus associated with DF-1 compared with the same inoculated DF-1 cells in cell-free fraction at (**A**) 6 hpi, (**B**) 12 hpi, and (**C**) 24 hpi. All data points (cts) were first converted to GE/mL, and then, the percentage of releases was calculated, according to the equation previously described. GE, genomic equivalent. One-way ANOVA with Tukey’s multiple comparisons post hoc test, *p* ≤ 0.05, ns; not statistically significant.

**Figure 8 viruses-14-01225-f008:**
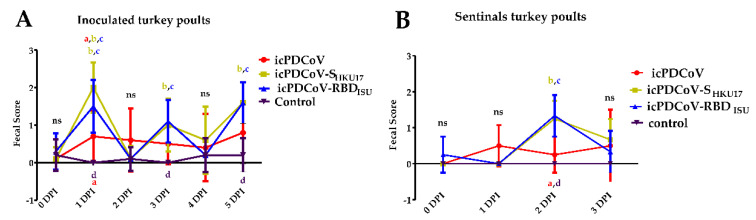
Fecal scores of turkey poults at 0- to 5-DPI. Fecal consistency was scored as follows: 0, normal; 1, pasty; 2, semiliquid; 3, watery, and the fecal score >2 was considered as diarrhea. Red circles, green squares, blue triangles, and black triangles represent (**A**) icPDCoV-inoculated, icPDCoV-S_HKU17_-inoculated, icPDCoV-RBD_ISU_-inoculated, and control groups (*n* = 10 biologically independent samples). (**B**) sentinel poults that commingled at 1 DPI (*n* = 4 biologically independent samples), respectively. Different lower-case letters (a; icPDCoV, b; icPDCoV-S_HKU17_, c; icPDCoV-RBD_ISU_, d; mock) indicate significant differences between the given group at each time point. Significant diarrhea for each test virus was only observed at day 1 post infection but for groups icPDCoV-S_HKU17_ and icPDCoV-RBD_ISU_ diarrhea was also observed on days 3 and 5 in inoculated birds and day 2 in sentinel birds. Two-way ANOVA with Bonferroni post hoc test, *p* ≤ 0.05. ns, not statistically significant.

**Figure 9 viruses-14-01225-f009:**
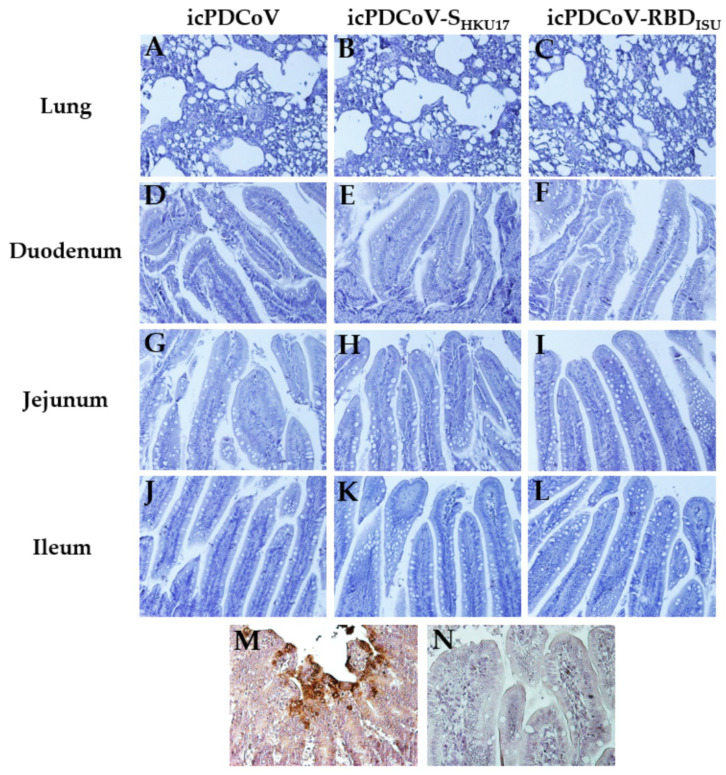
Immunohistochemistry was used to detect PDCoV NP in the lung, duodenum, jejunum, or ileum tissues from turkey poults. Birds were inoculated with- icPDCoV (**A**,**D**,**G**,**J**), icPDCoV-S_HKU17_ (**B**,**E**,**H**,**K**), or icPDCoV-RBD_ISU_ (**C**,**F**,**I**,**L**), showing no cellular signal for PDCoV N antigen. PDCoV-infected pig intestine used as positive (**M**) and negative (**N**) controls (magnification, 300×).

**Figure 10 viruses-14-01225-f010:**
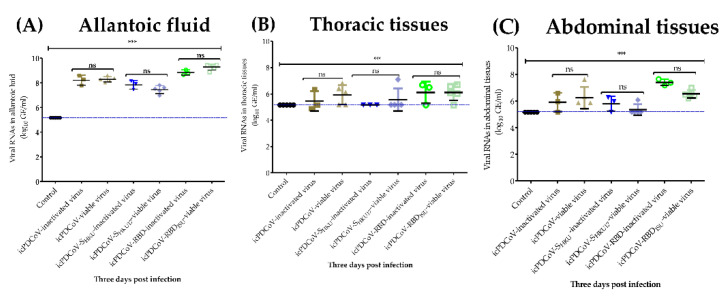
Quantity of viral RNAs in (**A**) allantoic fluids, (**B**) thoracic- and (**C**) abdominal tissues. The 11-day-old embryonated chicken eggs were inoculated with mock-inoculated, or heat-inactivated (at 60 °C for 20 min) or viable icPDCoV, icPDCoV-S_HKU17_, and icPDCoV-RBD_ISU_ viruses. Dashed line indicates detection limit of 5.17 log10 GE/mL of viruses in samples. GE, genomic equivalent. One-way ANOVA with Tukey’s multiple comparisons post hoc test, *** *p* ≤ 0.05, ns; not statistically significant.

**Table 1 viruses-14-01225-t001:** Replication of chimeric porcine deltacoronaviruses in 5-day-old embryonic chicken eggs.

	icPDCoV			icPDCoV-RBD_isu_			icPDCoV-S_HKU17_		
	ECE Mortality Rate	Cell Infectivity Positive Rate	RT-PCR	ECE Mortality Rate	Cell Infectivity Positive Rate	RT-PCR	ECE Mortality Rate	Cell Infectivity Positive Rate	RT-PCR
Original *	NT	NT	NT	NT	NT	NT	0(0/5)	60(3/5)	40(2/5)
10^−1^	NT	NT	NT	NT	NT	NT	0(0/5)	0(0/5)	NT
10^−2^	100(5/5)	100(5/5)	100(2/2)	50(3/6)	100(4/4)	100(2/2)	0(0/5)	0(0/5)	NT
10^−3^	100(5/5)	100(5/5)	NT	0(0/5)	60(3/5)	NT	NT	NT	NT
10^−4^	100(5/5)	100(5/5)	NT	40(2/5)	20(1/5)	NT	NT	NT	NT
10^−5^	100(5/5)	100(5/5)	NT	NT	NT	NT	NT	NT	NT
10^−6^	0(0/5)	20(1/5)	20(1/5)	NT	NT	NT	NT	NT	NT

NT = Not tested. *: The original virus stocks of icPDCoV, icPDCoV-RBD-isu and icPDCoV-S_KHU17_ had infectious titers of 7.8, 9.7 and 10.6 log_10_ median tissue culture infectious dose (TCID_50_/mL), respectively.

## Data Availability

The data presented in this study are available in this article and on request from the corresponding author.

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
