# Peer review of "Characterization of the Cross-Species Transmission Potential for Porcine Deltacoronaviruses Expressing Sparrow Coronavirus Spike Protein in Commercial Poultry"

_viruses, 2022, doi:10.3390/v14061225_

Round 1
Reviewer 1 Report
In my opinion, the manuscript is nicely written and is based on rigorous experimental work.
I have only technical comments.
Figure 5,6. Any abbreaviation should be deciphered. Please specify the mening of "CA" and "CF" in the Figures legend.
"viral RNA loads in DF-1 cells from virus associated 345
with DF-1 cells and cell-free fractions increased longitudinally with time (Figure 6A, B). 346"
Please compare viral RNA load between different time points. This suggestion seems to be incorrect.
Please clarify the methology of "Fecal scores" estimation in Materials and Methods section.
Please clarify the meaning and coloring of "lower-case letters (a, b, c, d)" at Figure 8. Overlapping colors make it impossible to evaluate the change of fecal scores.
Author Response
We would like to start by thanking the reviewers for their time and helpful critiques of this submission. We have gone through the requested edits and addressed each reviewer’s concerns with modification to the submitted manuscript. We will address each modification in a point-by-point response below.
Comments and Suggestions for Authors
Reviewer 1:
In my opinion, the manuscript is nicely written and is based on rigorous experimental work.
We thank the reviewer for their kind words. The finished article was a great collaborative work and I appreciate the hard work by the editors, researchers, students, and collaborators to make it a success.
I have only technical comments.
Q- Figure 5,6. Any abbreviation should be deciphered. Please specify the meaning of "CA" and "CF" in the Figures legend.
A- We have clarified that CA is an abbreviation of cell-associated and CF is an abbreviation of cell-free for each figure legend (5 and 6).
Q- "viral RNA loads in DF-1 cells from virus associated 345
with DF-1 cells and cell-free fractions increased longitudinally with time (Figure 6A, B). 346"
Please compare viral RNA load between different time points. This suggestion seems to be incorrect.
A-The reviewer is correct and we misstated the graph interpretation. A small increase in RNA for each virus between 6-12 hours was observed with no increase between 12-24 hours congruent with the TCID50 result presented in figure 4B. We have modified lines 344-347 as follows:
“…Figure 4B), our analyzed samples showed that viral RNA loads in DF-1 cells from virus associated with DF-1 cells and cell-free fractions slightly increased from 6-12 hours post infection but remained constant from 12-24 hours (Figure 6A, B).”
Q-Please clarify the methodology of "Fecal scores" estimation in Materials and Methods section.
A- We have made text revisions to line 205 -207 to help improve the methodology for determining fecal scores.
Q-Please clarify the meaning and coloring of "lower-case letters (a, b, c, d)" at Figure 8. Overlapping colors make it impossible to evaluate the change of fecal scores.
A-We have attempted to clarify this figure. The lower-case letters are meant to show statistical difference between groups using multivariate analysis via ANOVA. For instance, of panel 8A at 1DPI, birds infected with icPDCoV, icPDCoV-SHKU17 and icPDCoV-RBDisu (a, b, c) were all significantly different from each other, mock and a (icPDCoV) were significantly lower than groups b; icPDCoV-SHKU17 and c; icPDCoV-RBDISU. We have tried to clarify in the figure legend, but this might just be a case of overuse of statistical analysis. If it has not been significantly clarified, we can further alter the image for clarity.
Reviewer 2 Report
The authors performed additional experiments at new timepoints in order to further explore the replication capacity of the chimeric viruses.
only some minori points:
-Please describe TCID50 acronym in the text.
- Fig4: having CCIF for LLC-PK1 cells and TCID50 for DF-1 is dis-homogeneous. Could the authors add a panel with TCID50 for LLC-PK1?
-About my previous observation: Paragraph 2.8: the GE/mL is calculated on volume of swab supernatants? On eluted RNA? Please specify.
A-We have added information that specifies that the GE/mL is based upon the eluted RNA volume and can be directly compared as the swab supernatant volumes were normalized with one another.
Is not clear to me where the authors added the requested information.
Author Response
We would like to start by thanking the reviewers for their time and helpful critiques of this submission. We have gone through the requested edits and addressed each reviewer’s concerns with modification to the submitted manuscript. We will address each modification in a point-by-point response below.
Comments and Suggestions for Authors
Reviewer 2
Comments and Suggestions for Authors
The authors performed additional experiments at new timepoints in order to further explore the replication capacity of the chimeric viruses.
only some minor points:
Q-Please describe TCID50 acronym in the text.
A-We have inserted the median tissue culture infectious dose description of acronym TCID50
Q- Fig4: having CCIF for LLC-PK1 cells and TCID50 for DF-1 is dis-homogeneous. Could the authors add a panel with TCID50 for LLC-PK1?
A-We agree with the reviewer that showing CCIF for LLC-PK1 cells and TCID50 for DF-1 is dis-homogeneous. We have removed panel 4A (CCIF results) We have added previously generated data for TCID50 of icPDCoV performed in LLC-PK1 cells as a reference kinetic curve and referred readers to the Niu et al 2021 Viruses paper in which the infectious clone and chimeras were initially characterized in pigs and porcine cell lines.
Q-About my previous observation: Paragraph 2.8: the GE/mL is calculated on volume of swab supernatants? On eluted RNA? Please specify.
A-We have added information that specifies that the GE/mL is based upon the eluted RNA volume and can be directly compared as the swab supernatant volumes were normalized with one another prior to RNA extraction. We have clarified this in the text.
Q- Is not clear to me where the authors added the requested information.
- We do apologize for not sending a tracked changes version, there does not appear to be a specific file location for this type of document on the Viruses upload center. We will attempt to send a tracked version to the editor to distribute to the reviewer to more easily find the edits based upon their helpful comments.
A- We believe based on the further questions that you were reviewer number 2 for the previous submission. We have used bold text to highlight where the reviewer should expect to find the changes based upon their comments.
We have modified the figure legend to Figure 2 to denote DPC is day post comingling.
Why did the authors decide to euthanize all the 16 sentinels at day 3 and not at day 5 DPI?
A-Previous studies by Boley et al suggested that initial pathology in turkeys occurred early in infection and resolved rapidly. While PDCoV is typically self-limiting and self-resolving, we wanted to focus on the days we believed would present the most significant gross and microscopic pathological lesions. As this was a potentially novel host-virus interaction we likely should have performed pathology at varying time points but overlooked this experimental detail.
We have added additional information regarding our choice of day 5 to the discussion section.
Paragraph 2.8: the GE/mL is calculated on volume of swab supernatants? On eluted RNA? Please specify.
A-We have added information that specifies that the GE/mL is based upon the eluted RNA volume and can be directly compared as the swab supernatant volumes were normalized with one another.
Please check if ref 43 cited in line 432 is correct.
We replace the second instance of reference 43 with appropriate reference to CoV spike-mediated fusion.
Line 470-471: the cited reference [51] report a direct (not inverse) correlation between interferon response and viral load. In the paper the authors report “analyses showed a clear correlation between up-regulation of interferon-related genes and high viral load…high viral load might have induced strong IFN responses in the upper respiratory tract” and this is in line with a defense response against a high viral load.
A- The reviewer is correct that there is a direct correlation between interferon and viral load. We have modified the text in lines 470-471 to reflect this detail.
Line 477: the hypothesis related to the trypsin use is not clear to me, since it is used with both cell type, so it cannot be an explanation for the differences in replication found in the two cell lines.
A-We agree with the reviewer that trypsin concentration was likely not a primary cause for infectivity loss as the LLC-PK1-derived virus did not present similar loss of infection kinetics. We have removed this implausible explanation from the text. We have removed the CCIF data in figure 4A completely and changed the focus to kinetics in DF-1 cells utilizing the TCID50 data. We feel the sensitivity of the DF1 cells to trypsin might have led to premature cell loss leading to some of the aberrant data. Redoing the DF1 infection without trypsin retained the less infectious phenotype but did not rapidly lose infectious virions.
This manuscript is a resubmission of an earlier submission. The following is a list of the peer review reports and author responses from that submission.
Round 1
Reviewer 1 Report
This manuscript is evaluation of the chimera PDCoV virus with SpDCoV spike or RBD in cells, chicken embryonated eggs, and turkey poults for replication, pathogenesis and pathology. The data showed that spike and RBD chimeric PDCoV viruses replicate inefficiently in DF-1 cell and no replication in 11-day-old ECEs. Inoculated turkey poults showed undetectable virus RNA in both colacal and tracheal swabs. Overall, these data showed that the chimeric PDCoV is not pathogenic to avian in ovo and in vivo. Further understanding of these clones in vitro and in vivo is needed.
How many passage of these viruses in vitro were performed?
Reviewer 2 Report
In the Manuscript entitle “Characterization of the Cross-Species Transmission Potential for Porcine Deltacoronaviruses Expressing Sparrow Corona-virus Spike Protein in Commercial Poultry” the authors present a characterization of two recombinant chimeric viruses, expressing the S protein or the RBD of the sparrow CoV (SpCoV) respectively, on a backbone of an infectious clone of porcine CoV (icPDCoV).
The infectiveness and replicative potential was evaluated both in vitro and in vivo. In vitro experiments were performed on DF-1 chicken cells and LLC-PK1 porcine cells (used as reference). In vivo experiments were conducted on 6-days old turkey poults as well as on 11-days old embryonated chicken eggs.
In my opinion, the study is very well designed and explained. Please find below some comments for clarification.
Major point:
- There is only one major point that need to be clarified, regarding the starting clone icPDCoV. I understand the authors are already conscious about a lack of the genomic characterization of the clone that could evidences some main differences respect to the originating PDCoV. Anyway I was wondering if the authors at least performed an in vitro evaluation, comparing the infective and replication capacity of the used icPDCoV versus a PDCoV derived from pigs intestinal content. Actually, what is not clear to me is if the results obtained in LLC-PK1 by the icPDCoV infection are as expected or if the efficiency of infection should be higher. If the in vitro experiments are ok, I do not understand why the authors in line 498 wrote “The derivation of the icPDCoV sequence from virus passaged 8 times in tissue culture genomic mutations may have accrued rendering the cell passed virus less able to infect and cause pathology than the virus inoculum derived from intestinal contents of gnotobiotic pigs. Additional studies comparing infectivity of icPDCoV and intestinal content derived PDCoV along with virus sequence comparisons are needed to determine if specific mutations are responsible for a reduction of infectivity observed for the icPDCoV derived virus in inoculated turkey poults.”
Some minor points for clarification
- Figure 2.: please add the acronym explanation for DPC (day post commingling, I suppose).
- Why did the authors decide to euthanized all the 16 sentinels at day 3 and not at day 5 DPI?
- Paragraph 2.8: the GE/mL is calculated on volume of swab supernatants? On eluted RNA? Please specify.
- Please check if ref 43 cited in line 432 is correct.
- Line 470-471: the cited reference [51] report a direct (not inverse) correlation between interferon response and viral load. In the paper the authors report “analyses showed a clear correlation between up-regulation of interferon-related genes and high viral load…high viral load might have induced strong IFN responses in the upper respiratory tract” and this is in line with a defense response against a high viral load.
- Line 477: the hypothesis related to the trypsin use is not clear to me, since it is used with both cell type, so it cannot be an explanation for the differences in replication found in the two cell lines.
Reviewer 3 Report
In my opinion, the manuscript is nicely written and is based on rigorous experimental work.
I have only technical comments.
Figures 4-8, 10 has bad quality obstructing the perception of the material. Figure 5 has an unannotated white stripe ahead of the bars.
Please fix this.